# Elucidation of native California *Agave americana* and *Agave deserti* biofuel potential: Compositional analysis

**May Ling Lu**[1,2], **Charles E. Wyman**[1,2]*

1 Center for Environmental Research and Technology, Bourns College of Engineering, University of California, Riverside, California, United States of America, 2 Department of Chemical and Environmental Engineering, Bourns College of Engineering, University of California, Riverside, California, United States of America

☯ These authors contributed equally to this work.

\* cewyman@engr.ucr.edu

**Data Availability Statement:** All relevant data are within the manuscript and its Supporting Information files.

**Funding:** The project was funded in part by the Ford Motor Company, which established a chair in

## Abstract

Because biofuels have the unique potential to be rapidly deployed in existing transportation fuel infrastructures, they should play a major role in helping California quickly meet its aggressive goals to substantially reduce greenhouse gas contributions by this major sector. Furthermore, energy crops are vital to significantly impact the State's large and burgeoning need for sustainable fuels. Among crops amenable to be grown in California to support fuel production, agave pose a particularly promising prospect, given their drought tolerance and high productivity on marginal land in a State prone to drought and limited water resources. This study focuses on measuring compositional profiles of wild *A. deserti* and cultivated *A. americana*, two agaves native to California, to elucidate their potential for biological conversion to fuels that can help meet the huge State need for low-carbon transportation. Results from this study indicate that these two California agave species can be rich in fructans, ranging from 96–314 g/L of equivalent fructose and glucose in their leaf bases. In addition, structural and water-soluble sugar contents exceeding 63 wt.% show that these plants are amenable to fermentation to ethanol and other biofuels. Moreover, because the low K-lignin content of agave leaf bases bagasse of only about 12–18 wt.% suggests low recalcitrance and the negligible acid insoluble ash content should facilitate pretreatment prior to fermentations, the agave species native to the State hold considerable promise as potential biofuel feedstocks.

## Introduction

Biofuels constitute a critical means for satisfying California's enormous gasoline appetite while simultaneously facilitating its progressive march towards the creation of a greener and more sustainable energy future. In addition, because many biofuels are compatible with the existing transportation infrastructure, they can avoid the time lost to fleet turnovers required for many low carbon options to power transportation and thereby have a much more immediate effect

Environmental Engineering at the Center for Environmental Research and Technology (CE_CERT), Bourns College of Engineeering at the University of California, Riverside (UCR), for Charles E. Wyman (CEW). In addition, a 2014 University of California Center on Economic Competitiveness in Transportation (UCCONNECT) (Award No.: 00008347) graduate fellowship was awarded to May Ling Lu (MLL) to support her effort on agave compositional analysis. Link to the UCCONNECT website can be found at http://ucconnect.berkeley.edu/. Finally, the authors are grateful for the provision of the 2014 UCR Graduate Fellowship by the UCR Bourns College of Engineering to May Ling Lu (MLL), which also facilitated this study. All the funders had no role in study design, data collection and analysis, decision to publish, or preparation of the manuscript.

**Competing interests:** We have read the journal's policy and the authors of this manuscript have the following competing interest: The Ford Motor Company, which funded a chair in Environmental Engineering at the Center for Environmental Research and Technology (CE_CERT), Bourns College of Engineeering at the University of California, Riverside (UCR) for Charles E. Wyman. This funding source, however, does not alter our adherence to PLOS ONE policies on sharing data and materials.

on greenhouse gas emissions from the transportation system. Currently, the State imports corn ethanol from the Midwest and sugarcane-based ethanol from Brazil, the latter to help meet the requirements of the Low Carbon Fuel Standard [1, 2] that targets progressive reductions in carbon intensity of transportation fuels in California [1]. With the goal of 20% reduction in carbon intensity by 2030 [1], the use of biofuels from lignocellulosic biomass becomes more appealing as these biofuels can provide greater reductions in GHG accumulations than corn or sugarcane derived ethanol [3]. Because dedicated lignocellulosic energy crops can be more abundant than available biomass residues [4], they are important to include if the State is to sustainably satisfy its large demand for transportation fuels.

For successful commercial application of energy crops, high productivity is required. In California, such productivity should be accomplished on marginal areas not suitable for food crop cultivation, especially arid and semi-arid lands receiving no more than 18 inches of rain [5]. These regions, located southwest of the State, cover Riverside, San Bernandino and Imperial Counties, based on the current Köppen-Geiger Climate Classification system, which maps the global vegetation distribution according to climate gradients [6]. This feature is desirable as the State is prone to drought [7, 8] and subjected to tight water resources due to its burgeoning population [7]. Moreover, bioenergy crops priced at $60/ton [4] and with productivity similar to California maize of approximately 9 tons/acre of total biomass of grain and stover [9] would have values of less than $600/acre, which makes it prohibitive for them to compete with California's irrigated land crops which can have values exceeding $10,000/acre [10].

Fortunately, there are some plant species in the State that possess the desirable characteristics of high productivity and low water requirement. Agave plants, in particular, are adapted to areas of low rainfall due to their outstanding water use efficiency (WUE). For some agave species, their WUE can be six times greater than the more dominant C3 plants [5]. For example, *A. deserti* in the Sonoron Desert of California, where the annual rainfall is about 430 mm (17 inches), is highly productive with yields of 7 dry metric tons/ha/year [11]. This productivity is considered attractive compared to many other energy crops and particularly promising for desert areas characterized by average yields in the low one metric ton/ha/year, and where individual life forms such as lichens have productivity of less than 0.1 metric ton/ha/year [12].

Though agave has appealing characteristics, few studies have focused on California native agave species and their potential for biofuel conversion. Ample studies were made of the more popular non-native *A. tequilana* due to the abundant residues of leaves and bagasse leftover following agave harvesting and tequila production, respectively [13–18]. Though *A. americana*, a potential native species of the State, was investigated by Corbin et al. [15] and Rijal et al. [17] their agave species were from Australia. Gonzalez-Llanes et al. [19] also studied the reducing sugars of *A. americana* leaves in their enzymatic hydrolysis of mezcal-producing agaves, but their samples were sited in Mexico. In the United States, Li et al. [16] established the composition of 4–5 years old *A. americana* that was sourced from San Jose, California, and found the plant to be a promising feedstock candidate as its unpretreated agave leaf and stem bagasse when subjected to enzymatic hydrolysis at enzyme loading of 150 mg/g of structural carbohydrates resulted in high total sugar yields exceeding 70 wt.% [20]. Nobel [5, 12] generated research on the native *A. deserti* and its adaptability to and productivity in the desert, but did not establish the chemical compositional profile of the plant.

The restriction to native species minimizes the threat posed by non-native plants becoming invasive when introduced into the fauna and flora of the State [21]. *A. americana*, for instance, is classified as an invasive plant in parts of Australia [22] and Macronesia [23] and was listed as an environmental weed that threatened biodiversity in the former [24]. Though mass scale cultivation of the native species can be a challenge, especially if they are not commercial species, native agave species have the advantage of being well-suited to the local climatic and edaphic

conditions, which may render the plants more resilient against pests and diseases and less reliant on fertilizers [21, 25]. Both a cultivated, nursery-based 3–4 year old *A. americana* and an older wild *A. deserti* from the desert area were selected in this study to represent the diverse spectrum of conditions from which the native species can be found. Thus, this study will assess the potential of California native agaves from contrasting backgrounds as energy crops by elucidating their chemical composition for biofuel production to meet the State's biofuel need.

## Materials and methods

### Materials

Only four agave species are native to California: *A. americana*, *A. deserti*, *A. shawii*, and *A. utahensis* [26]. This study focused only on the first two as both *A. utahensis* and *A. shawii* fell short of the target characteristics of a plant amendable to commercialization. *A. utahensis* is a small agave plant [27] found in the Desert Mountains (i.e., Clark and Ivanpah Mountains and Kingston Range [28]) that lacks the potential for large scale conversion into biofuels. *A. shawii*, on the other hand, is adapted to coastal regions [29] with higher moisture requirement, which diminishes its appeal as a drought-tolerant plant. Though the native status of *A. americana* is debatable as the Jepson Herbarium database, which focuses on the vascular plants of California, does not list it as such [30]; nevertheless, it is included in this study as the plant has shown significant potential as a bioenergy crop. Previous research showed that *A. americana* was able to yield more than 50 wt.% sugar without pretreatment at enzyme loading of 15 mg/g of structural carbohydrate after 6 days of hydrolysis and at a higher enzyme loading of 150 mg/g, the actual sugar yield approached at least 65 wt.% after only 3 days [20]. *A. deserti*, as noted, has high productivity (7 dry metric tons/ha/year) in semi-arid conditions [11]. Consequently, both these species were included in this study.

The samples obtained were a whole plant harvested from the Pinyon Flat in the Santa Rosa Mountains, for *A. deserti* and a cultivated 3–4 years old *A. americana* from the Desert Theater Nursery in Escondido, San Diego, California. The wild *A. deserti* was provided courtesy of Mr. Daniel McCarthy, the former Director of Cultural Resources Management Department of the San Manuel Band of Mission Indians. It was a mature, medium-sized plant that weighed 19.3 kg (fresh weight) without dead leaves and roots and was estimated to be about 20 years old by Mr. Daniel McCarthy (personal communication, June 28, 2015). *A. americana*, on the other hand, was smaller and weighed around 4.7 kg with the roots and dead leaves removed. Since little stem is available from *A. americana* by observation and as reported in literature [27, 31], the leaf base (see Fig 1) attached to the stem that is higher in complex sugars content of fructo-oligomers than the tip were analyzed in place of the stem [19, 32]. For both species, although the leaf apex was not included at this stage, a more comprehensive assessment of the plant would be desirable if results for the leaf bases show commercial promise.

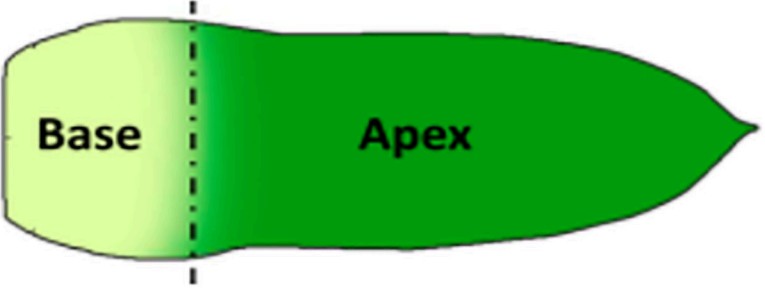

**Fig 1. Leaf base (light hue) and apex (green) of agave plant** [33].

## Methods

The leaf base was obtained by first cutting the individual leaf of the plant with a 33 cm (13 inch) curved saw tooth knife (Corona RS 7120, Corona Clipper, Inc., CA), which left behind a 13–15 cm (5–6 inches) protrusion from the stem. This action facilitated the prying of the "initial" leaf base from its rosette-like structure around the stem. The leaf bases, leaf apex, and stems were brushed to remove dirt, stored in the freezer at -4˚C, and thawed when needed.

Following the first cut, the initial leaf bases still have an amount of green leaf apex attached. These were removed with a smaller and more wieldable *Cutco* serrated knife (CUTCO Model 3738C, CUTCO, NY) that sliced through the fibrous leaf with greater precision. The remaining leaf bases, which constituted the resource used throughout the study, were then diced into smaller cubes (0.64 cm (1/4 inch) in dimension) for easier handling. Mass difference between the agave species meant that all the leaf bases for *A. americana* were utilized while only half as many were used for *A. deserti*. The leftover leaf apex was stored in the freezer for future analysis.

**Compositional analysis of agave juice.** Juice was expelled by squeezing the leaf bases cubes against a flat base filter at the bottom of a customized 24.1 cm (9.5 inch) long by 8.9 cm (3.5 inch) diameter metal cylinder using a tight fitting piston powered by a manually operated 12 ton hydraulic press (Model No. 14 590, Northern Tool + Equipment, Burnsville, MN) [20]. Following extraction, the turbid agave bagasse juice containing solid and small debris was pipetted into 50 ml polypropylene tubes for centrifuging at 3400 rpm for 25 minutes (Allegra® X-15R, Beckman Coulter Inc., Brea, CA). The resulting visually clear supernatants from all tubes were mixed together before dispensation into smaller aliquots of 5–10 ml for storage in a -4˚C freezer. This centrifuged juice was utilized throughout the study.

The juice was analyzed for its free sugars content of sucrose, fructose, glucose, arabinose, and galactose using a Waters Alliance e2695 HPLC, equipped with a Phenomenex Rezex™ RPM-Monosaccharide Pb+2 (Torrance, CA) column and a refractive index (RI) detector (Waters Corporation, Milford, MA). The Waters HPLC operated at 0.6 ml/min of double deionized water and at temperature of 75˚C.

Fructans, oligomers and polymers of fructose with a single glucose moiety in the juice were first hydrolyzed before HPLC analysis by adapting the downscaled method [16] devised by DeMartini, Studer, and Wyman [34] that followed the National Renewable Energy Lab (NREL) post-hydrolysis method for hydrolysate oligomers conversion to monomers [35]. However, modifications of the NREL procedure were necessary since the more labile fructose was generated in fructans hydrolysis. Thus, instead of utilizing 4 wt.% sulfuric acid at 121˚C for 1 hour of reaction time [35], milder reaction of 2 wt.% sulfuric acid at 105˚C for 1 hour was found to improve sugar quantification by minimizing fructose degradation. Fructose degradation was about 90% at the original NREL conditions [36] while application of the milder conditions resulted in approximately 25–30% degradation. Sugar recovery standards were prepared. Due to the high concentration of fructose (>80 g/L) from fructan hydrolysis, the juice was diluted by a factor of 10 before reaction. Since fructans were converted to their basic fructose and glucose units in hydrolysis and analyzed in the HPLC as such, they are expressed in equivalent fructose and glucose concentrations (g/L) in the study. This value, however, over-represented the actual fructan content due to the reaction of water molecules with the polydisperse fructans in hydrolysis. Quadruplicate samples were made for both the free sugars and the downscale fructan content evaluation.

**Compositional analysis of agave bagasse.** The leftover from hydraulic pressing was analyzed for water soluble carbohydrates (WSC) and extractives, structural carbohydrates, lignin,

and ash contents. Prior to compositional analysis, the bagasse was lyophilized for 24–48 hours to less than 10 wt.% moisture in a freeze-dryer (FreeZone® 4.5 Liter Freeze Dry Systems, Labconco, Kansas City, MO), knife-milled to ≤ 2 mm in a Thomas Wiley® mill (Model 4, Thomas Scientific, Swedesboro, NJ), and then milled further through a 40 mesh screen in a mini mill (Model No. 3383-L20. Thomas Scientific, Swedesboro, NJ) to facilitate homogenization of the particle size for analysis [16]. To determine the WSC, a 1g dry sample was soaked in 15 ml of deionized water and incubated in a shaker (Multitron HT Infors, ATR Biotech, Laurel, MD) at 50°C for 24 hours and 150 rpm [16]. Sodium azide was also added to the solution at concentration of 0.2 g/L to avert microbial contamination. Following incubation, the content was centrifuged for 20 mins at 3,200 rpm (Allegra X-15R, Beckman Coulter, Brea, CA) to effect separation of the bagasse. The resulting supernatant was analyzed for both free sugar monomers and fructans according to the procedure applied to juice analysis. Samples were analyzed in quadruplicate.

Measurement of bagasse extractives was according to the NREL procedure with a Soxhlet apparatus [37]. Water-based extraction in duplicates was performed using roughly 5 g of sample per extraction. The solution containing the extractives was evaporated in an isotemp vacuum oven (Model 281A, Fisher Scientific, Dubuque, IA). For structural carbohydrates, lignin, and acid-insoluble ash content analysis, the NREL procedure involving application of concentrated sulfuric acid (72 wt.%), followed by dilute sulfuric acid hydrolysis (4 wt.%), was applied to quadruplicates of water soluble extractives-free biomass [38]. Whole ash determination was performed according to NREL method on ash content [39]. Fig 2 shows a flowchart summarizing the juice and biomass components analyzed and the methods for quantifying the individual components.

## Results and discussion

Table 1 highlights the composition of the leaf bases of *A. americana* and *A. deserti* in terms of the structural components of carbohydrates, ash, and lignin content. WSC and extractives contents complete the information developed for bagasse. WSC, similar to juice, was comprised of free sugars and fructans. Note that in this study, where fructans were involved, they are expressed in equivalent fructose and glucose concentration (g/L). Consequently, the polysaccharide content of WSC and the juice shown in Table 1 can be greater in value than their actual content, which was devoid of hydrolyzed sugars. For example, *A. americana* WSC, expressed in wt.% of equivalent fructose and glucose (g) per amount of dry biomass substrate (g) at 31.0 wt.%, exceeded its water-soluble extractives content of 30.4 wt.%, which was determined as amount of extractives (g) recovered from solvent evaporation per amount of dry biomass substrate (g). Detailed breakdown of the structural carbohydrates and fructans into their monomeric components is provided in Tables 2 and 3.

As indicated in Table 1, *A. americana* leaf bases have a low structural carbohydrate content of 39.4% compared to *A. deserti*, with its leaf bases structural carbohydrate content of 54.7%. Table 2, which delineates the breakdown of the polysaccharides, points to a low glucan and xylan content for the 3–4 year old *A. americana*, which were 22% and 8.0%, respectively. Though low, they fell within the range published in the literature. Li et al. [16], for instance, reported 4–5 year old *A. americana* leaves as possessing roughly 30% glucan and 8% xylan. Corbin et al. [15], on the other hand, showed an even lower glucose and xylose content of 12.0 and 2.9%, respectively, (or 10.8% and 2.5% glucan and xylan content) for her 2–3 year *A. americana* leaves. Thus, the 3–4 year old *A. americana* leaf bases with glucan and xylan contents totaling ~30% were within the literature range of 13–38%.

When coupled with WSC, the total carbohydrate contents of *A. americana* and *A. deserti* leaf bases were 70.4% and 63.2%, respectively. Although these amounts rivaled the sugar

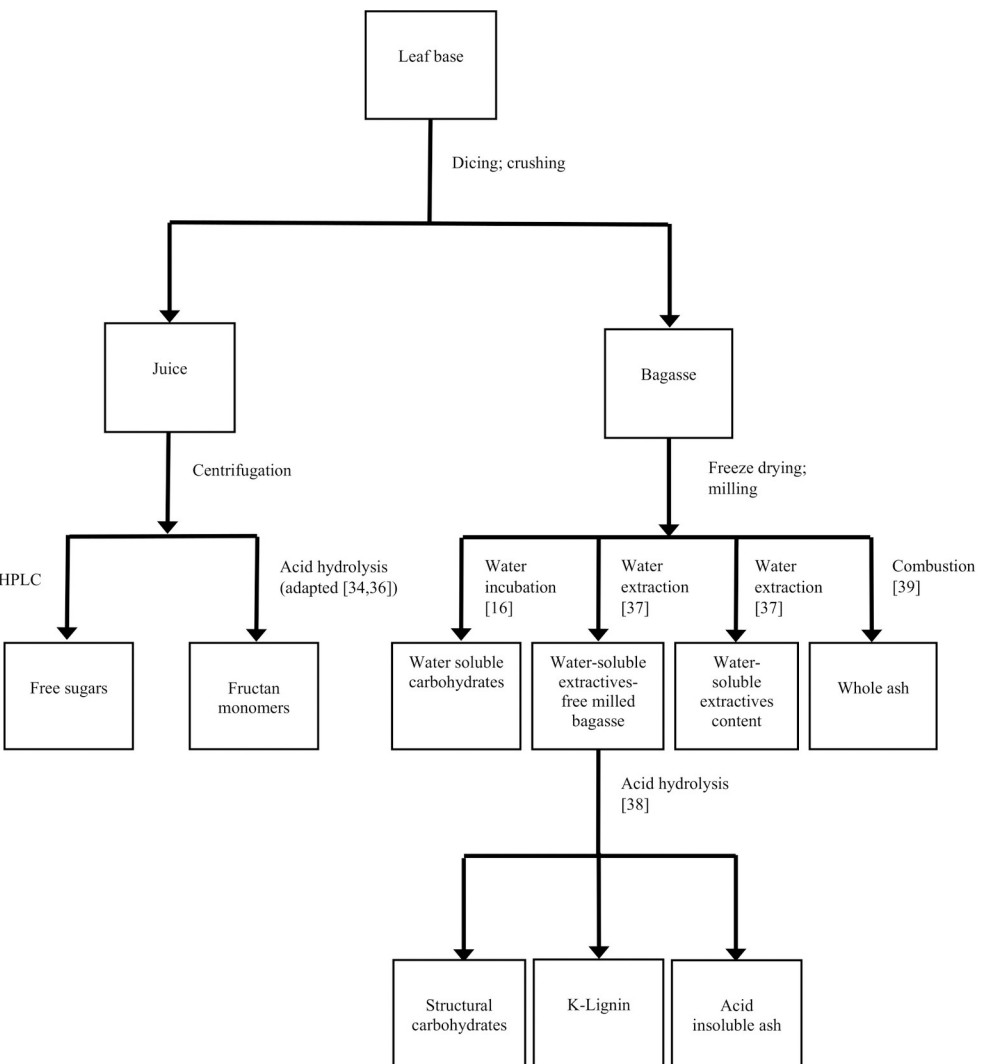

**Fig 2. Flowchart summarizing juice and bagasse components analyzed and methods used.**

content of hardwood poplar, with an average combined cellulosic and hemicellulosic composition of 64.8% [40], and exceeded those of the ubiquitous corn stover and the energy crop switchgrass by about 4–11% [40], it should be noted that the total combined carbohydrate in Table 1 includes the hydrolyzed fructans from the WSC and were derived from the sugar-rich leaf bases [19]. With declining soluble sugar content away from the leaf base [19], the WSC would decline when the whole leaf was analyzed. Nevertheless, when the WSC, which constitutes a significant portion of available carbohydrate associated with the bagasse [16], was factored into consideration, agave continues to look promising as a potential biofuel feedstock.

The polysaccharides in the bagasse of *A. americana* leaf bases contained primarily C6 sugar units, with xylan and arabinan constituting no more than 24.4% of the total structural carbohydrate. With *A. deserti*, the C5 content was almost a third of the structural sugar content at 31.1%. This fact is important in fermentations in that the traditional industrial yeast workhorse *S. cerevisiae* can readily metabolize monosaccharides and disaccharide of glucose, fructose, and sucrose [41], but not C5 sugars [42, 43].

**Table 1. Composition analysis of juice and dry bagasse of California *A. americana* and *A. deserti* leaf bases.**

| | Bagasse (As-received) | | | | | | | | | | | | |
|---|---|---|---|---|---|---|---|---|---|---|---|---|---|
| Sample Type | Structural Carbohydrate[a] | | K-Lignin[a] | | Water Soluble Carbohydrate[a,b] (WSC) | | Water-soluble Extractives Content[c] | | Acid Insoluble Ash[a] | | Whole Ash[a] | | Total[a,b,d] |
| | Avg. | SE | Avg. | SE | Avg. | SE | Avg. | SE | Avg. | SE | Avg. | SE | Avg. |
| | wt.% | wt.% | wt.% | wt.% | wt.% | wt.% | wt.% | wt.% | wt.% | wt.% | wt.% | wt.% | wt.% |
| *A. americana* | 39.4 | 0.9 | 11.6 | 0.2 | 31.0 | 0.2 | 30.4 | 0.5 | nd | | 4.5 | 0.0 | 85.9 |
| *A. deserti* | 54.7 | 0.7 | 17.5 | 0.4 | 8.5 | 0.1 | 11.8 | 0.0 | nd | | 3.6 | 0.0 | 87.6 |
| | Juice | | | | | | | | | | | | |
| | Free Sugars[a] | | Fructans[a,e] | | Total | | | | | | | | |
| | Avg. | SE | Avg. | SE | Avg. | SE | | | | | | | |
| | g/L | g/L | g/L | g/L | g/L | g/L | | | | | | | |
| *A. americana* | 22.6 | 0.4 | 313.9 | 1.5 | 336.5 | 1.9 | | | | | | | |
| *A. deserti* | 34.9 | 0.1 | 95.9 | 0.7 | 130.7 | 0.8 | | | | | | | |

*Note.* Avg. = Average. SE = Standard Error. wt.% = fraction (by weight) of dry biomass

[a]Based on quadruplicates.

[b]WSC is expressed in equivalent fructose and glucose per dry bagasse as received.

[c]Based on duplicates.

[d]The total covers structural carbohydrates, water-soluble extractives content, K-lignin, and whole ash.

[e]Fructans are expressed as equivalent glucose and fructose (g/L).

The lignin content, measured as acid insoluble K-lignin, was high for both species compared to literature reports. The 11.6% lignin content of *A. americana* used in this study was greater than that measured by Li et al. for a 4–5 year old leaf sample (8.2 wt.%) [16] and Corbin et al. for a 2–3 year old leaf sample (5.3 wt.%).[15] It was also lower than corn stover lignin content of 13–14% [40, 44] and those of energy crops, such as switchgrass (16.1%) and hybrid poplar (24.9%) [40]. The K-lignin content of *A. deserti* approached 18.0%. Because a low lignin content suggests less plant recalcitrance, *A. americana* is expected to be more amendable to bioethanol conversion.

Acid-insoluble ash was not detectable in either *A. americana* or *A. deserti* leaf bases. A whole ash composition in the range of 3–5% is typical of a wide variety of agave species, with ash content varying from 1–7% [45]. Increased ash content could be detrimental to pretreatment processes utilizing dilute sulfuric acid in the processing of biomass to biofuels in that the neutralizing capacity of the alkali metal components in ash could reduce the effectiveness of the acid catalyst in the pretreatment step [40].

The total bagasse composition analyzed, which consists of structural carbohydrates, K-lignin, water-soluble extractives content, and acid insoluble ash constitute up to ~86 and

**Table 2. Breakdown of structural carbohydrates in California *A. americana* and *A. deserti* leaf bases bagasse.**

| | Bagasse Structural Carbohydrate Composition | | | | | | | | | | | |
|---|---|---|---|---|---|---|---|---|---|---|---|---|
| Sample Type | Glucan | | Xylan | | Galactan | | Mannan | | Arabinan | | Total | |
| | Avg. | SE | Avg. | SE | Avg. | SE | Avg. | SE | Avg. | SE | Avg. | SE |
| | wt.% | wt.% | wt.% | wt.% | wt.% | wt.% | wt.% | wt.% | wt.% | wt.% | wt.% | wt.% |
| *A. americana* | 22.0% | 0.5% | 8.0% | 0.2% | 6.7% | 0.3% | 1.2% | 0.1% | 1.6% | 0.0% | 39.4% | 0.9% |
| *A. deserti* | 30.1% | 0.4% | 15.2% | 0.2% | 6.3% | 0.1% | 1.2% | 0.0% | 1.8% | 0.0% | 54.7% | 0.7% |

*Note.* Avg. = Average. SE = Standard error based on quadruplicates. Compositions were measured on dry weight basis.

**Table 3. Breakdown of free sugars and fructans in California *A. americana* and *A. deserti* leaf bases juice.**

| | Free Sugars and Fructan Compositions of Agave Juices | | | | | | | | | | | |
| --- | --- | --- | --- | --- | --- | --- | --- | --- | --- | --- | --- | --- |
| | Glucose | | Fructose | | Sucrose | | Galactose | | Arabinose | | Total | |
| | Avg. | SE | Avg. | SE | Avg. | SE | Avg. | SE | Avg. | SE | Avg. | SE |
| | g/L | g/L | g/L | g/L | g/L | g/L | g/L | g/L | g/L | g/L | g/L | g/L |
| *A. americana* | | | | | | | | | | | | |
| Free Sugar | 7.6 | 0.1 | 7.3 | 0.2 | 6.4 | 0.1 | 0.8 | 0.0 | 0.5 | 0.0 | 22.6 | 0.4 |
| Fructan[a] | 23.3 | 0.2 | 290.6 | 1.6 | | | | | | | 313.9 | 1.5 |
| *A. deserti* | | | | | | | | | | | | |
| Free Sugar | 17.1 | 0.0 | 10.8 | 0.0 | 5.9 | 0.0 | 0.6 | 0.0 | 0.5 | 0.0 | 34.9 | 0.1 |
| Fructan[a] | 22.4 | 0.1 | 73.5 | 0.8 | | | | | | | 95.9 | 0.7 |

*Note*. Avg. = Average. SE = Standard error based on quadruplicates.

[a]Fructan concentration is expressed as g/L of equivalent fructose and glucose.

88% of the dry leaf bases bagasse of *A. americana* and *A. deserti*, respectively. The remaining fraction could potentially include protein [16], acid-soluble lignin, acetate groups, and lipids and waxes according to Corbin et al. [15].

The structural composition of the two plants differed notably in their lignin and carbohydrate content. One possible explanation is their wide age gap. The wild *A. deserti*, which was estimated to be 20 years old, has a higher lignin content befitting an older agave plant with greater number of lignin dominant fibers in the leaves. As studied by Smith and Nobel, the number of vascular bundles in *A. deserti*, which is proportional to the leaf surface area [46], could be higher in the older and larger sample *A. deserti*, whose aboveground leaves fresh weight of 8.5 kg was distributed among 33 leaves while that of *A. americana* was 2.1 kg and distributed among 16 leaves. In addition, Oudiani et al. showed that plants with older leaves in the outer level contain fibers dominated by lignin in contrast to the younger leaves found in the inner positions that are rich in hydrogen bonding among the cellulose comprising the fibers [47]. In their study of *A. sisalana* fibers extracted from leaves ranging in ages from 2–9 years, Chand and Hashmi concluded that lignin content was maximum at the oldest age [48]. Owing to its age, *A. deserti* contained potentially more fibrous bundles with greater lignin content.

The higher structural carbohydrate content in the older *A. deserti* could have resulted from the preponderance of non-lignified and thin walled parenchyma cells associated with water storage tissues found in agave leaves as observed by Li et al. using the confocal laser-scanning microscopy [20]. Singh et al.'s terahertz imaging of succulent *A. vistoriae-reginae* leaves indicated the base to be high in water content and therefore, concentrated with hydrenchyma cells [49]. Bernandino-Nicanor et al. noted that hydrenchyma cells in *A. atroviren* Karw increased with age from 3 to 9 years old to better provide support for the plant during drought [50]. The greater presence of these cells to protect against hydric stress [50] provides one possible explanation for the higher structural carbohydrates of older *A. deserti* leaf bases relative to that of *A. americana*.

Both species are rich in fructan concentration in the juice. *A. americana*, in particular, with a fructan concentration of 313.9 g/L of equivalent fructose and glucose, has three times higher concentration than *A. deserti*, with a concentration of 95.9 g/L (see Table 3). Such a high fructan concentration not only adds significantly to ethanol production potential but greatly enhances agave as a biorefinery feedstock as fructan is a versatile compound that can be converted into various value added products, especially in the nutritional or food sectors. They included pre-biotics, a beneficial bacterial stimulant of the human gut microflora; a

substitute for fat in food, such as yogurts, spreads, and ice-cream; and low calorie sweeteners [51, 52].

Agave plants show wide variation in their sugar compositions due to species, age, environmental influences, and agronomic practices. Borland et al. noted that agave cultivars differed significantly in their sugar and fructan content [53]. Reports on *agave* stems and basal leaves where reserve carbohydrates are stored consistently show fructan content increased from the younger to the older plants in agave species, such as cultivated *A. tequilana* [15, 54–56] and *A. salmiana* [57]. This trend held among wild *A. tequilana*, *A. salmiana*, *A. mapisaga*, and *A. atrovirens*, as studied by Aldrete-Herrera et al. [58], who saw a roughly 10% increase in fructan portion of reducing sugars from the 2–4 years age group to the 10–12 years age group. This result is in keeping with the function of the agave stem as a storage organ for the significant energy required for its reproduction [20, 57] at the end of its life where a towering stalk, along with thousands of flowers and seeds are produced [59]. This trend, however, was not observed for the much older leaf bases of wild *A. deserti* vs. the 3–4 year old cultivated *A. americana*. The lower fructan contents of *A. deserti* could be affected not only by species and age differences, but by the environment and agronomic practices, which favor the latter. Environmental influences, in the form of climate and edaphic conditions, were found to impact carbohydrate profile of *A. tequilana* sourced from two regions in Mexico [60]. Though genetically identical the regional influences caused the water soluble carbohydrate from the sample stem of *A. tequilana* of Jalisco to almost double that of Guanajuato [60]. Furthermore, as noted by Davis, Dohleman, and Long, agronomic practices of pruning, water irrigation, weeding, and fertilizer and pesticide applications enhanced productivity [61]. Nobel in his study of *A. tequilana*'s and *A. deserti*'s total soluble carbohydrate content in both the leaves and the stem indicated that the variable responded to changes in the environmental productivity index (EPI), a measure of plant yield [62]. With proper management, total soluble carbohydrate concentration in both leaves and stem rose as this variable tracked plant productivity. Given that the *A. americana* whole sample plant was furnished by a San Diego nursery that performed plant maintenance in contrast to the *A. deserti* sample plant, which was harvested from the wild of the Santa Rosa Mountain, it is plausible for the fructan concentration of a younger plant to significantly exceed that for a much older plant.

The free sugar contents of *A. americana* and *A. deserti* juice, comprised of glucose, fructose, and sucrose, were 21.3 g/L and 33.8 g/L, respectively. Assuming that other sugar monomers (e.g., arabinose, mannose, galactose) were present in minor to negligible amounts, the work by Gonzalez-Llanes et al. [19] indicated a leaf base reducing sugar concentration of 19.7 g/L for their 8 year old *A. americana* harvested from Mexico. Given that the leaf base was similar in soluble carbohydrate content to the stem [19], these compositions were also comparable to stem juice compositions from other studies. In their article, Li et al. [16] noted that their 4–5 year old *A. americana* stem sample from San Jose had 27.4 g/L of free sucrose, fructose and glucose combined, which were higher than the values obtained in this study for the same species. The concentrations of 21.3 g/L and 33.8 g/L constitute roughly 6% and 26% of the total soluble saccharides in the juice of *A. americana* and *A. deserti*, respectively. The presence of high monosaccharides of fructose and glucose and disaccharide sucrose concentration should be an advantage to biological biofuel production technologies as such sugars require little energy or enzyme input to deconstruct them to fermentable sugars prior to conversion by *S. cerevisiae*.

## Conclusions

The California agave species *A. americana* and *A. deserti* utilized in this case study are drought tolerant plants with low water requirements that have high potential as biorefinery

feedstocks, especially in a state that is prone to precipitation shortages but burgeoning water demand. Although the two differed respectively in bagasse structural carbohydrate composition (39.4 wt.% for *A. americana* vs. 54.7 wt.% for *A. deserti*), juice fructan and free sugar contents (336.5 g/L vs. 130.7 g/L), and K-lignin composition (11.6 wt.% vs. 17.5 wt.%) possibly due to species, age, geographic variations, and cultivation, they demonstrated potential as energy crops for biofuel conversion. Whether cultivated as for the *A. americana* evaluated or harvested from the wild as for the *A. deserti* used here, both species are rich in fructans that can be converted to fermentable fructose and glucose. Their fructan concentrations are high at 314 g/L for *A. americana* juice and 96 g/L for *A. deserti* juice. Their bagasse structural carbohydrates, combined with water soluble carbohydrates (expressed in equivalent fructose and glucose units), are at least 63.2%, exceeding those of the widely studied switchgrass and corn stover by about 4–11%. Moreover, among energy crops, California *A. americana* stacks up well against hybrid poplar woods and switchgrass not only in its high sugar content (inclusive of fructans) but also in its reduced recalcitrance by way of its lower lignin content at 11.6 wt.%, which cements its appeal as a competitive biorefinery feedstock candidate.

## Supporting information

**S1 File. Supporting information for compositional analysis of juice and dry bagasse of California *A. americana* and *A. deserti* leaf bases in Tables 1–3.** The file contains data for computation and statistical analysis of structural carbohydrates, water soluble carbohydrates (WSC), water-soluble extractives-free contents, K-lignin, acid insoluble ash, and whole ash composition of *A. americana* and *A. deserti* leaf bases bagasse, and free sugars and fructans content of their juice.
(XLSX)

## Acknowledgments

Special thanks to the Center for Environmental Research and Technology, Bourns College of Engineering at the University of California, Riverside, for providing facilities to conduct this research and to Mr. Daniel McCarthy, the former Director of Cultural Resources Management Department of the San Manuel Band of Mission Indians, for graciously providing the wild *A. deserti* and sharing his observations of the plant.

## Author Contributions

**Conceptualization:** Charles E. Wyman.

**Investigation:** May Ling Lu.

**Methodology:** May Ling Lu.

**Project administration:** May Ling Lu.

**Resources:** Charles E. Wyman.

**Supervision:** Charles E. Wyman.

**Writing – original draft:** May Ling Lu.

**Writing – review & editing:** Charles E. Wyman.

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
