## [Decision Letter · Decision Letter 0]

8 Feb 2021

PONE-D-20-41108

Elucidation of native California Agave americana and Agave deserti biofuel potential: compositional analysis

PLOS ONE

Dear Dr. Wyman,

Thank you for submitting your manuscript to PLOS ONE. After careful consideration, we feel that it has merit but does not fully meet PLOS ONE’s publication criteria as it currently stands. Therefore, we invite you to submit a revised version of the manuscript that addresses the points raised during the review process.

Please note that one of the reviewers made several valuable suggestions and raised some questions that you should address before your manuscript can be accepted for publication. I would kindly recommend you to also address the concern made by reviewer #2 regarding technical/biological replicates.

We look forward to receiving your revised manuscript.

Kind regards,

Igor Cesarino, Ph.D

Academic Editor

PLOS ONE

Journal Requirements:

Reviewers' comments:

Reviewer's Responses to Questions

**Comments to the Author**

1. Is the manuscript technically sound, and do the data support the conclusions?

Reviewer #1: Yes

Reviewer #2: Yes

2. Has the statistical analysis been performed appropriately and rigorously? 

Reviewer #1: Yes

Reviewer #2: I Don't Know

3. Have the authors made all data underlying the findings in their manuscript fully available?

Reviewer #1: Yes

Reviewer #2: Yes

4. Is the manuscript presented in an intelligible fashion and written in standard English?

Reviewer #1: Yes

Reviewer #2: Yes

5. Review Comments to the Author

Reviewer #1: Comments to the authors (PONE-D-20-41108). The manuscript by Lu and Wyman reports detailed compositional analysis of two promising but understudied Agave species (deserti and americana) which due to their drought tolerance capabilities including biomass high productivity makes it attractive as another promising bioenergy feedstock.

In my consideration, the study was well-performed, conclusions are sustained by research data and in general, the study is relevant for the Plos One journal.

Nevertheless, this manuscript needs clarification and improvements and clarification in order to be considered for publication in the following sections.

+General comments:

*Which is the land feasible for Agave cultivation and/or arid/semi-arid land in California?

*Provide an example of the species with this low biomass productivity (Page 10, line 52).

*The A. deserti sample must include the approximate cultivation age in the methods section, as an older sample would definitely have different characteristics that a younger sample.

*It is not appropriate to compare the composition of bagasse vs. leaves as it has different components. There are a few published papers that report the lignin content of Agave americana bagasse (i.e. Yang et al., 2012. Bioresource Technology, 126, 336) for your consideration.

*The compositional analysis of the bagasses in Table 1, are in dry basis? If the composition sums 86-88% then what comprises the rest (protein? wax?)?

*One issue is that this Agave americana bagasse is different from the one reported elsewhere, where they are cooked for Mezcal production where typically plants aged 7-8 years are used.

*Figure 1 is missing (Page 12, line 110).

*It is suggested to include a flowchart of the compositional analysis methods employed.

In brief, the current form of this manuscript cannot be accepted, but the authors are encouraged to perform modifications based on the above comments to modify their manuscript and submit a revised manuscript.

Reviewer #2: Very interesting comparative analysis of the biomass properties of wild (A. deserti) versus cultivated (A. americana). Methods seem appropriate and are adequately explained. Results indicate that Agave varieties such as A. deserti have the potential to contribute genetic factors to improve Agave breeding for biomass quantity/quality. My only major concern is with regard to the number of individual plants assayed (1 each, as far as I can tell). Statical analysis was presented as standard error therefore represent technical error (not biological replicates) by my estimation. It would be better have analysed samples from multiple individual plants (biological replicates) to better evaluate natural variation among ecotypes. Nonetheless, the conclusions were within reason and are generally supported by their data.

6. PLOS authors have the option to publish the peer review history of their article (what does this mean?). If published, this will include your full peer review and any attached files.

Reviewer #1: No

Reviewer #2: No

---

## [Author Response · Author response to Decision Letter 0]

26 Apr 2021

March 26, 2021

Dr. Igor Cesarino

Academic Editor

PLOS ONE

Subject: Response to reviews and revisions for PLOS ONE manuscript PONE-D-20-41108 entitled “Elucidation of native California Agave americana and Agave deserti biofuel potential: compositional analysis” by May Ling Lu and and Charles E. Wyman

Dear Dr. Cesarino:

We are grateful to the reviewers for their thoughtful reviews of the manuscript entitled “Elucidation of native California Agave americana and Agave deserti biofuel potential: compositional analysis” by May Ling Lu and Charles E. Wyman and for their suggestions that enhance the lucidity and technical merits of our paper. Below we provide point-by-point responses to each of the comments made by the reviewers and indicate how the manuscript was changed to address each one. 

Reviewer #1:

Comment: *Which is the land feasible for Agave cultivation and/or arid/semi-arid land in California?

Response: Arid and semi-arid regions in California include those located in the southwestern part of the State (i.e., San Bernandino, Riverside, and Imperial Counties) according to the Köppen-Geiger Climate Classification system, which mapped the global vegetation distribution according to climate gradients. This information was added to the manuscript Introduction section (p. 3, Lines 39-41). 

Comment: *Provide an example of the species with this low biomass productivity (Page 10, line 52)

Response: According to the reference paper from which the biomass productivity is obtained, one metric ton/ha/yr (Page 4, line 52-54) is an average figure. The manuscript Introduction section (page 4, Line 54-56) has been revised to incorporate lichen, with productivity of 0.1 metric ton/ha/yr, to demonstrate the slow growth rate of some other desert life forms. 

Comment: *The A. deserti sample must include the approximate cultivation age in the methods section, as an older sample would definitely have different characteristics that a younger sample.

Response: Thank you for detecting the omission. The Material section of the manuscript (Page 6, Line 105-106) has been revised to include the estimated age of the plant and the Results and Discussion section (Page 16, Line 255-257) has been also modified to reflect the adjustment. 

Comment: *It is not appropriate to compare the composition of bagasse vs. leaves as it has different components. There are a few published papers that report the lignin content of Agave americana bagasse (i.e. Yang et al., 2012. Bioresource Technology, 126, 336) for your consideration.

Response: The term “bagasse” as used in the manuscript refers to the fibrous/pith portion of the leaf base after the juice is extracted. We are unsure how to respond to this comment as the leaf base “bagasse” structural composition and lignin content in the Results and Discussion section of this manuscript were compared to those of other leaves as reported by Li et al. and Corbin et al. (Page 14, Line 212-218 for structural carbohydrate; Page 15, Line 235-240 for K-lignin). Please advise on how to proceed if we are not interpreting the Reviewer’s comments correctly. 

Comment: *The compositional analysis of the bagasses in Table 1, are in dry basis? If the composition sums 86-88% then what comprises the rest (protein? wax?)?

Response: Yes, the compositional analysis of the leaf base bagasse is on a dry basis. The total composition is the sum of water-based extractive, whole ash, K-lignin, and structural carbohydrates. Literature analysis indicates that the remaining unknown can be made up of proteins, acid-soluble lignin, acetate groups, and lipids and waxes. This explanation has been added to the Results and Discussion section (Page 15, 249-253) and an explanation of the sum of the composition has also been added to the notes of Table 1 (Page 11). 

Comment: *One issue is that this Agave americana bagasse is different from the one reported elsewhere, where they are cooked for Mezcal production where typically plants aged 7-8 years are used.

Response: We did not make changes to the manuscript to address this comment as we are not clear of the specific differences between the A. americana leaf base bagasse reported here and the one to which Reviewer #1 was referencing. But generally speaking, the A. americana bagasse composition used in mezcal production is expected to be different from the one analyzed in this manuscript. The explanation describing the age factor and its impact on structural carbohydrates and lignin contents differences between A. deserti and A. americana in the manuscript (Page 16, lines 254-276) would apply to this particular comparison between the A. americana studied (3-4 yrs) and the one used in mezcal production (7-8 years). One would also anticipate differences due to the processing (cooking and milling) that A. americana bagasse was subjected to during mezcal production vs. the raw leaf base bagasse analyzed. 

Comment: *Figure 1 is missing (Page 12, line 110).

Response: Thank you for pointing this out. We will upload Fig 1 (Page 6, Line 109), which highlights an agave leaf showing a partition between the leaf base and the leaf apex.

Comment: *It is suggested to include a flowchart of the compositional analysis methods employed.

Response: Thank you. A flow chart has been added as Fig 2 (Page 10, Line 186-188)

Reviewer #2

Comment: Very interesting comparative analysis of the biomass properties of wild (A. deserti) versus cultivated (A. americana). Methods seem appropriate and are adequately explained. Results indicate that Agave varieties such as A. deserti have the potential to contribute genetic factors to improve Agave breeding for biomass quantity/quality. My only major concern is with regard to the number of individual plants assayed (1 each, as far as I can tell). Statical analysis was presented as standard error therefore represent technical error (not biological replicates) by my estimation. It would be better have analyzed samples from multiple individual plants (biological replicates) to better evaluate natural variation among ecotypes. Nonetheless, the conclusions were within reason and are generally supported by their data.

Response: Thank you for reviewing our manuscript and the thoughtful feedback. You are correct in that only one sample was used as the source for our data. The analysis is a case study that would add to the overall picture on the promise of agave as a potential biofuel feedstock. It provides an exploratory snapshot of California wild and cultivated agave and their potential albeit constrained by local conditions at which the agave species were grown. Further more extensive work, especially multiple samples, would be needed to delve into the natural variation of a species within its ecosystem. Such an investigation, however, would require a very large number of samples such as have been used to understand corn stover variability to be meaningful and is beyond the scope of our intent and objective to explore the potential of agave species native to California. Consequently, biological replicates were not taken. 

We also made minor changes as highlighted by the tracking changes in the paper to further improve the paper. 

Again, we thank you and the reviewers for the thoughtful suggestions and the opportunity to modify our manuscript for PLOS ONE consideration. We hope these responses and changes in the paper properly address these thoughtful points but would be pleased to offer additional clarification as needed.

Sincerely,

Charles E. Wyman, PhD, MBA

Distinguished Professor, Department of Chemical and Environmental Engineering, 

Ford Motor Company Chair in Environmental Engineering in the

Bourns College of Engineering Center for Environmental Research and Technology, and

Founding Editor-in-Chief of Biotechnology for Biofuels

---

## [Decision Letter · Decision Letter 1]

12 May 2021

Elucidation of native California Agave americana and Agave deserti biofuel potential: Compositional analysis

PONE-D-20-41108R1

Dear Dr. Wyman,

We’re pleased to inform you that your manuscript has been judged scientifically suitable for publication and will be formally accepted for publication once it meets all outstanding technical requirements.

One important suggestion has been made by reviewer #2. As your manuscript is presented as a 'case-study', the reviewer suggests the addition of a short (1-sentence) statement acknowledging this somewhere in your conclusions.

Kind regards,

Igor Cesarino, Ph.D

Academic Editor

PLOS ONE

Additional Editor Comments (optional):

Reviewers' comments:

Reviewer's Responses to Questions

**Comments to the Author**

1. If the authors have adequately addressed your comments raised in a previous round of review and you feel that this manuscript is now acceptable for publication, you may indicate that here to bypass the “Comments to the Author” section, enter your conflict of interest statement in the “Confidential to Editor” section, and submit your "Accept" recommendation.

Reviewer #1: All comments have been addressed

Reviewer #2: All comments have been addressed

2. Is the manuscript technically sound, and do the data support the conclusions?

Reviewer #1: Yes

Reviewer #2: Yes

3. Has the statistical analysis been performed appropriately and rigorously? 

Reviewer #1: Yes

Reviewer #2: N/A

4. Have the authors made all data underlying the findings in their manuscript fully available?

Reviewer #1: Yes

Reviewer #2: Yes

5. Is the manuscript presented in an intelligible fashion and written in standard English?

Reviewer #1: Yes

Reviewer #2: Yes

6. Review Comments to the Author

Reviewer #1: The authors have already answered appropriately all comments made by this reviewer, hence, it can be considered for publication.

Reviewer #2: As this manuscript is being presented essentially as a 'case-study', my previous concerns have generally been addressed. I agree...further analyses using multiple biological replicates will be necessary, but are indeed outside of the scope of this manuscript. A statement in the conclusion indicating such would be appreciated.

7. PLOS authors have the option to publish the peer review history of their article (what does this mean?). If published, this will include your full peer review and any attached files.

Reviewer #1: **Yes: **Jose A Perez-Pimienta

Reviewer #2: No

---

## [Editor Report · Acceptance letter]

19 May 2021

PONE-D-20-41108R1 

Elucidation of native California *Agave americana* and *Agave deserti* biofuel potential: compositional analysis 

Dear Dr. Wyman:

I'm pleased to inform you that your manuscript has been deemed suitable for publication in PLOS ONE. Congratulations! Your manuscript is now with our production department. 

Kind regards, 

on behalf of

Dr. Igor Cesarino 

Academic Editor

PLOS ONE